# Predicting mental health problems in adolescence using machine learning techniques

**Ashley E. Tate**[1]*, **Ryan C. McCabe**[2], **Henrik Larsson**[1,3], **Sebastian Lundström**[4,5], **Paul Lichtenstein**[1], **Ralf Kuja-Halkola**[1]

1 Department of Medical Epidemiology and Biostatics, Karolinska Institutet, Stockholm, Sweden, 2 Spotify, Stockholm, Sweden, 3 School of Medical Sciences, Örebro University, Örebro, Sweden, 4 Centre for Ethics, Law and Mental Health (CELAM), University of Gothenburg, Gothenburg, Sweden, 5 Gillberg Neuropsychiatry Centre, University of Gothenburg, Gothenburg, Sweden

* Ashley.thompson@ki.se

## Abstract

**Data Availability Statement:** We regret to say that we are unable to share even de-identified data, as legally bound by the Swedish Serecy Act. Data from the national Swedish registers and twin

### Background

Predicting which children will go on to develop mental health symptoms as adolescents is critical for early intervention and preventing future, severe negative outcomes. Although many aspects of a child's life, personality, and symptoms have been flagged as indicators, there is currently no model created to screen the general population for the risk of developing mental health problems. Additionally, the advent of machine learning techniques represents an exciting way to potentially improve upon the standard prediction modelling technique, logistic regression. Therefore, we aimed to I.) develop a model that can predict mental health problems in mid-adolescence II.) investigate if machine learning techniques (random forest, support vector machines, neural network, and XGBoost) will outperform logistic regression.

### Methods

In 7,638 twins from the Child and Adolescent Twin Study in Sweden we used 474 predictors derived from parental report and register data. The outcome, mental health problems, was determined by the Strengths and Difficulties Questionnaire. Model performance was determined by the area under the receiver operating characteristic curve (AUC).

### Results

Although model performance varied somewhat, the confidence interval overlapped for each model indicating non-significant superiority for the random forest model (AUC = 0.739, 95% CI 0.708–0.769), followed closely by support vector machines (AUC = 0.735, 95% CI 0.707–0.764).

register were used for this study and made available by ethical approval. The data used for this study include: Swedish Twin Registry, National Patient Register, Multi-Generation Register, Medical Birth Register, Prescribed Drug Register, the Longitudinal Integration Database for Health Insurance and Labor Market Studies. Researchers may apply for access these data sources through the Swedish Research Ethics Boards (etikprovningsmyndigheten.se; kansli@cepn.se) and from the primary data owners: Swedish Twin Registry (str-research@meb.ki.se), Statistics Sweden (scb@scb.se), and the National Board of Health and Welfare (socialstyrelsen@socialstyrelsen.se), in accordance with Swedish law.

**Funding:** The Child and Adolescent Twin Study in Sweden study was supported by the Swedish Council for Working Life, funds under the ALF agreement, the Söderström Königska Foundation and the Swedish Research Council (Medicine, Humanities and Social Science; grant number 2017-02552, and SIMSAM). SL, PL This work has received funding from the European Union's Horizon 2020 Research and Innovation Programme under the Marie Sklodowska-Curie CAPICE Project grant agreement number 721567. (https://www.capice-project.eu/) AT, PL, SL We acknowledge financial support from the Swedish Research Council for Health, Working Life and Welfare (project 2012-1678; PL), the Swedish Research Council (2016-01989; PL), as well as the the Swedish Initiative for Research on Microdata in the Social And Medical Sciences (SIMSAM) framework (340-2013-5867; PL). We acknowledge The Swedish Twin Registry for access to data. The Swedish Twin Registry is managed by Karolinska Institutet and receives funding through the Swedish Research Council under the grant no 2017-00641. The funders had no role in study design, data collection and analysis, decision to publish, or preparation of the manuscript.

**Competing interests:** I have read the journal's policy and the authors of this manuscript have the following competing interests: H. Larsson has served as a speaker for Evolan Pharmaand Shire and has received research grants from Shire; all outside the submitted work. P. Lichtenstein has served as a speaker for Medice, also outside the submitted work. R. McCabe serves as a data scientist for Spotify outside of the submitted work. All other authors declare that no competing interests exist. This does not alter our adherence to PLOS ONE policies on sharing data and materials.

## Conclusion

Ultimately, our top performing model would not be suitable for clinical use, however it lays important groundwork for future models seeking to predict general mental health outcomes. Future studies should make use of parent-rated assessments when possible. Additionally, it may not be necessary for similar studies to forgo logistic regression in favor of other more complex methods.

## Introduction

Childhood onset psychopathology can carry a heavy burden of negative outcomes that persist through adolescence and into adulthood. These outcomes are often severe: criminal convictions, low educational attainment, unemployment, and increased risk of suicide attempts [1, 2]. As many of the documented risk factors for mental illnesses in adolescence can be mitigated by early interventions [3], research establishing the most informative mental health indicators could help more precisely identify the proper traits for intervention targets.

There are several well researched indicators in childhood that are associated with the development of mental health problems. Psychopathological traits in early childhood also often indicate a higher risk for consistent mental health problems in adolescence and adulthood [4]; with even subthreshold symptoms indicating future adversity and a general predisposition to mental illnesses [5–7]. Internalizing and externalizing symptoms in childhood are both frequently associated with higher risk of mental illness diagnosis later in life [5, 8]. Specifically, impulsivity has been associated with a susceptibility of developing mental illnesses and suicide [9, 10]. Moreover, neurodevelopmental disorders, such as autism or ADHD, indicate lifelong diagnosis and frequent psychiatric comorbidities [11]. Similarly, learning difficulties can also indicate future mental health adversity and are frequently seen in children with neurodevelopmental disorders [12, 13].

Additionally, parental mental health, such as anxiety or depression, has been found to correlate with childhood internalizing and externalizing symptoms, likely due to a shared biologic (genetic) etiology[14, 15]. Thus, parental mental health may serve as an indicator of a more general predisposition for mental illness in lieu of genetic data. Genetic etiology is important to account for as most childhood psychiatric disorders overlap at both the phenotypic and etiological level [15]. Similarly, living in a lower SES neighborhood has been associated with an increase in internalizing problems and ADHD, although the mechanisms of this association are debated [16, 17]. Factors associated with the neonatal environment and birth have been associated with later adverse mental health and neurodevelopmental disorders [18, 19]. Moreover, chronic physical illness or disability can have a profound effect on mental health [20].

Taken together, reported factors in childhood associated with adolescent mental illness reflect intricate developmental pathways at almost every level. Understandably, most studies have not properly integrated risk factors from varying domains. Modern advancements in prediction modeling with machine learning may, in part, provide a cost-efficient solution to this problem.

### Machine learning in mental health

Supervised machine learning, used for classification or prediction modelling, has the advantage of accounting for complex relationships between variables that may not have been

previously identified. Thus, as datasets become larger and the variables more complex, machine learning techniques may become a useful tool within psychiatry to properly disentangle variables associated with outcomes for patients[21, 22].

A majority of studies using machine learning within psychiatry have focused on classification or diagnosis [23, 24]. However, critique has been raised that these studies are prone to under-perform due to a lack of insight on underlying assumptions of the various machine learning techniques or on the psychiatric disorders and corresponding diagnostic processes [25]; highlighting the difficulty in creating and validating such models. That said, advancements have been made in the field using tree based models to predict suicide in adolescents and the U.S. military [26, 27]. Beyond their proven efficacy, tree based models provide information on how extensively a variable was used for the model, or variable importance, which gives some insight to the models' classification process. This indicates that, while the way forward is arduous, properly conducted machine learning techniques can be interpretable and improve the efficacy of clinical decision making.

The primary aim for this study is to develop a model that can predict mental health problems in mid-adolescence. Additionally, we aim to investigate various machine learning techniques along with standard logistic regression to determine which performs best using combined questionnaire and register data. We expect that the techniques used will perform with similar accuracy according to the "No Free Lunch Theorem" [28, 29].

## Methods

### Participants

Participants came from the Child and Adolescent Twin Study in Sweden (CATSS), an ongoing, longitudinal study containing 15,156 twin pairs born in Sweden. During the first wave, the twins' parents were contacted close to their 9th or 12th birthdays for a phone interview, this wave had a response rate of 80% [30], while the second wave at age 15 had a response rate of ~55%. This sample population was chosen due to the depth of information available, including questionnaire and register data. Using the unique identification number given to all Swedes we linked several Swedish national registries to the CATSS data; the National Patient Register (NPR) [31], the Multi-Generation Register (for identification of parents) [32], the Medical Birth Register (MBR) [33], the Prescribed Drug Register (PDR) [34], as well the Longitudinal Integration Database for Health Insurance and Labor Market Studies (LISA) [35]. A total of 7,638 participants born between 1994 and 1999 who completed data collection at age 9 or 12 and again at age 15 were eligible for inclusion and were used in the analysis.

The study was approved by the Regional Ethical Review Board in Stockholm (the CATSS study, Dnr 02–289, 03–672, 2010/597-31/1, 2009/739-31/5, 2010/1410-31/1, 2015/1947-31/4; linkage to national registers, Dnr 2013/862–31/5).

### Measures

The outcome measure of adolescent mental health problems was collected at age 15 via the Strengths and Difficulties Questionnaire (SDQ) [36]. We used the SDQ to obtain parent-rated emotional symptoms, conduct problems, prosocial behavior, hyperactivity/inattention, and peer relationship problems. A binary variable was created based on a combination of the parent reported subscales, not including prosocial behavior, with a cut-off score validated for the Swedish population, corresponding to approximately 10% scoring above cut-off and thus rated as having mental health problems [37]. Predictors were collected at 9/12 or earlier from questionnaires administered through CATSS and through registers. We included a wide range

**Table 1. Information on techniques.**

| Technique | R Package used* | Descrption |
|---|---|---|
| Random Forest | RandomForest [51] | Decision trees are a model type that groups data in a tree like structure based on if-then-else decisions. At each decision point (node), data is branched off into smaller subgroups based on one of the predictor variables. Random forest is a method based on aggregating the results of many decision trees and prediction is determined based on the majority decision [52] |
| XGBoost | XGBoost [53] | XGBoost, or extreme gradient boosting, uses gradient boosting within random forest. Gradient boosting works by assigning scores to each leaf of the tree and builds new trees based on the performance of previously created trees, thus varying weight is assigned to each tree. This is in contrast to standard boosting techniques in random forest that work by assigning equal weights to trees [53]. |
| Logistic Regression | Base R | Logistic regression represents the standard method in epidemiology for analyzing binary outcomes [54]. In this model predictors are assumed to have a linear relationship to the outcome on the log-odds scale. Each predictor in the model has an associated regression coefficient which describes the direction and strength of its relationship to the outcome. We tested this model with interactions for all variables with sex, as well as with linear and quadratic effects for the A-TAC variables. |
| Neural Network | Neuralnet [55] | Neural network features numerous interconnected processors, or "neurons", organized in multiple layers: input, hidden, and output [55]. While there is only one input and output layer, there can be numerous hidden layers. During the learning process the input neurons respond to the data while neurons in the hidden and output layer respond to weighted connections from neurons at the previous layer. These weighted connections may be linear or non-linear and vary in complexity depending on the data and task [55]. Before analysis with this method, the predictors were scaled and centered. |
| Support Vector Machines | e1701[56] | Support Vector Machines works by dividing classes, i.e., cases versus non-cases, based on a line called a hyperplane. The hyperplane is created based on the greatest possible distance of the nearest neighboring predictor data points between the classes. Data with higher complexity that cannot be separated in two dimensions can be lifted to a higher dimension through a process called kernelling [57]. |

*mlr [42] was also used for all techniques

of predictors based on previous findings of association with adolescent mental health outcomes and/or childhood mental health. Predictors encompassed everything from birth information, physical illness, to mental health symptoms, to environmental factors such as neighborhood and parental income. Informants included both register and parental reported information. A total of 474 variables were initially included in the dataset, a complete list can be found in S1 File.

## Data pre-processing

Variables with more than 50% missingness were removed from analysis (202 variables excluded). Redundant variables were also removed (134 excluded). Additionally, variables with no variance were removed (32 excluded) and those with variance near zero were combined into one variable if possible, e.g. dust, mold, and pollen allergy collapsed into allergy [38]. Ultimately, 85 variables were determined to be suitable for analysis. As most machine learning techniques require complete datasets, missing values were imputed with tree based imputation with the R package mice [39].

## Statistical analysis

All analyses were performed in R. First, a learning curve was plotted with the entire dataset in order to check if our study was sufficiently powered.

Then, we split our data into a training-set (60% of the sample), a tune-set (10%), and a test-set (30%). Splitting data allows for more accurate determination in how the model will perform in a new dataset and helps alleviate overfitting, i.e. to fit the training data too closely to accurately predict other datasets. Stratified random sampling was used to ensure that the twin pairs would not get separated between the datasets, thus avoiding potential overfitting. Additionally, we preserved an equal distribution of the outcome between each set. Descriptive statistics were created for each set to determine the quality of the partition (Table 2).

We artificially inflated the number of cases in the training-set through a Synthetic Minority Over-sampling Technique (SMOTE), as implemented in the R-package SMOTEBoost [40], because positive cases were relatively rare. This phenomenon is commonly termed class imbalance [41] and can cause the model to predict all outcomes as the majority class.

The performance of predictions from considered models were determined by the area under the receiver operating characteristic curve (AUC). We created prediction models using several machine learning techniques: random forest, XGBoost, logistic regression, neural network and support vector machines (Table 1) to determine which produced the best fitting model for a test set. Using cross validation, each technique trained multiple models using the training set and tested their performance on a subset of the training set. The model with the lowest error was then tested using the tune set. Once the performance in the tune set was deemed satisfactory, the final models were then fitted to the test set. Parameter tuning was guided in part by standard practice when available, however a majority of the tuning took place through the random search function in R package mlr [42, 43]. Random search was completed using cross-validation with 3 iterations, 50 times. Variable importance was calculated for tree-based models: random forest and XGBoost. Confidence intervals at 95% were created for each AUC by bootstrapping predictions 10,000 times. Positive Predictive and Negative predictive values were obtained for the best performing model.

## Sensitivity analysis

The SDQ, used to derive our outcome variable, has several suggested cut-offs based on different criteria and sample populations. Although we used a cut-off of 11, based on capturing the highest 10% in a Swedish sample [37], it's possible that this cut-off does not represent a distinct subgroup of psychopathology, ultimately hampering model performance. To assess whether model performance was affected based on used cut-off, we created a new model using the best performing technique with a more stringent cut-off from the original publication. This cut-off of 17 was based on capturing the highest 10% of scorers in a UK sample in the original publication [36].

**Table 2.  Descriptive information from the partitioned data.**

|          | N    | Birth year   | Sex      | SDQ Cutoff        |
|----------|------|--------------|----------|-------------------|
|          |      | Mean (SD)    | Male %   | Cut off reached % |
| Trainset | 4554 | 1996.5 (1.69) | 48.4%   | 12.1%             |
| Tuneset  | 804  | 1996.3 (1.68) | 49.6%   | 12.3%             |
| Testset  | 2280 | 1996.5 (1.68) | 48.1%   | 11.5%             |

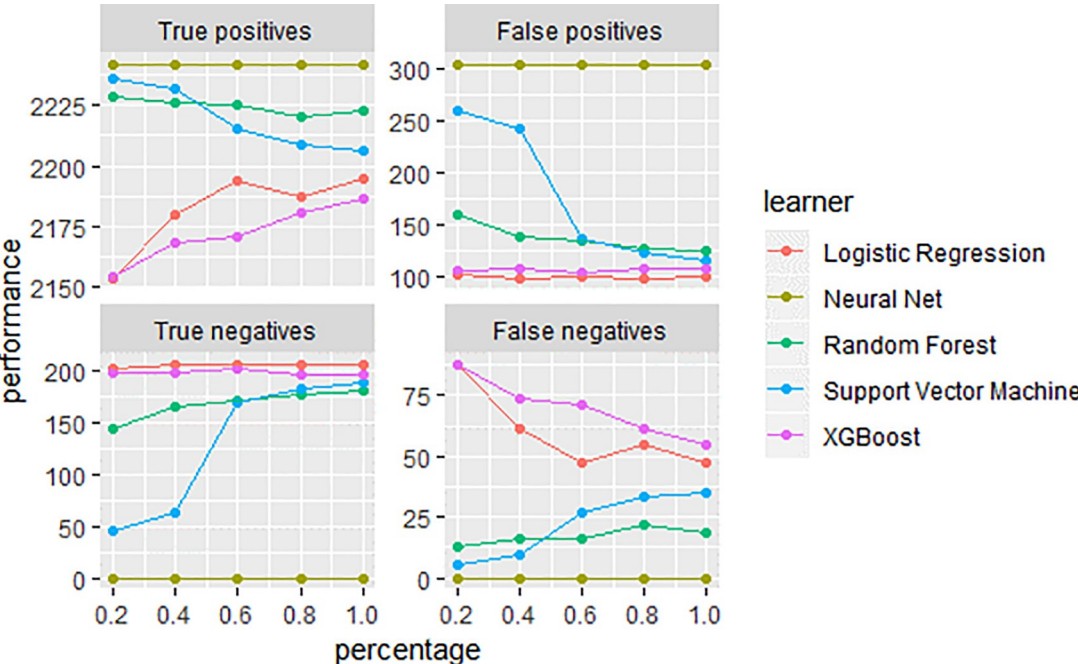

**Fig 1. Learning curve.** The learning curve specifying the performance of each technique without any data nor hyper-parameter modification (y axis) given the total percent of the dataset (x axis) used to train the models.

## Results

The datasets were deemed to be well separated (Table 2). Our classes were fairly imbalanced as only 12% of our sample reached the cut off, we mitigated the effects of this through a combination of over- and under sampling on the training set using SMOTEBoost. Next, the learning curve revealed that the models performed well without additional data nor hyper-parameter modifications, with an exception of neural network which required additional data preparation, e.g. centering and scaling of continuous variables (Fig 1).

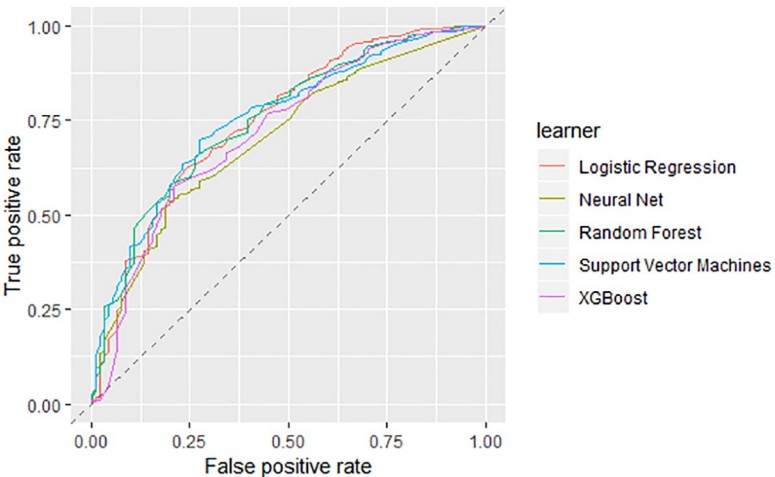

**Fig 2. AUC curves for tune set.** The AUC performance for each technique using the tune set.

Table 3. Model performance on tune set.

| Learner | AUC | 95% bootstrap interval |
|---|---|---|
| Logistic Regression | 0.750 | 0.693–0.805 |
| XGBoost | 0.723 | 0.662–0.778 |
| Random Forest | 0.754 | 0.698–0.804 |
| Support Vector Machines | 0.754 | 0.701–0.802 |
| Neural Network | 0.715 | 0.658–0.769 |

## Model tuning

We then fit models using all considered techniques; the AUCs from the tune-set of the final models for each technique can be found in Fig 2. A full list of the optimal parameters and the ranges tried for each model can be found in S1–S4 Tables. No model was found to be significantly superior, however random forest and support vector machine (SVM) had the highest AUCs of 0.754 (95% CI 0.698–0.804; and 95% CI 0.701–0.802, respectively). The rest of the models performed similarly with an AUC above 0.700 (Fig 2 & Table 3).

## Prediction

The created models were then used to predict the outcome in the test set. The lack of a statistically significant better model remained. The random forest model preformed slightly better at predicting the test set than SVM, with an AUC of 0.739 (95% CI 0.708–0.769) and 0.735 (95% CI 0.707–0.764) respectively (Table 4 & Fig 3), however the CI of each AUC overlaps the estimate of the other indicating a non-significant difference.

The probability threshold was set to 0.8, meaning that the model classified participants as having mental health problems if the probability of belonging to the class was greater than 0.2. Our top model had a predictive value of 15%, while the negative predictive value was at 96%. This corresponds to a sensitivity of .91 and a specificity of .30, and classified 15% of the test set with the outcome.

## Sensitivity analysis

The more stringent cut-off based on a UK sample [36] categorized roughly 3% of our sample as having mental health issues. We trained a random forest model based on this new cut-off, and found a test AUC of 0.765 (95% CI 0.698–0.826). Although the AUC was marginally better, the confidence interval overlapped with the top performing model with the Swedish cut offs, indicating no meaningful difference.

Table 4. Model performance on test set.

| Learner | AUC | 95% bootstrap interval |
|---|---|---|
| Logistic Regression | 0.700 | 0.665–0–734 |
| XGBoost | 0.692 | 0.660–0.723 |
| Random Forest | 0.739 | 0.708–0.769 |
| Support Vector Machines | 0.736 | 0.707–0.765 |
| Neural Network | 0.705 | 0.671–0.737 |

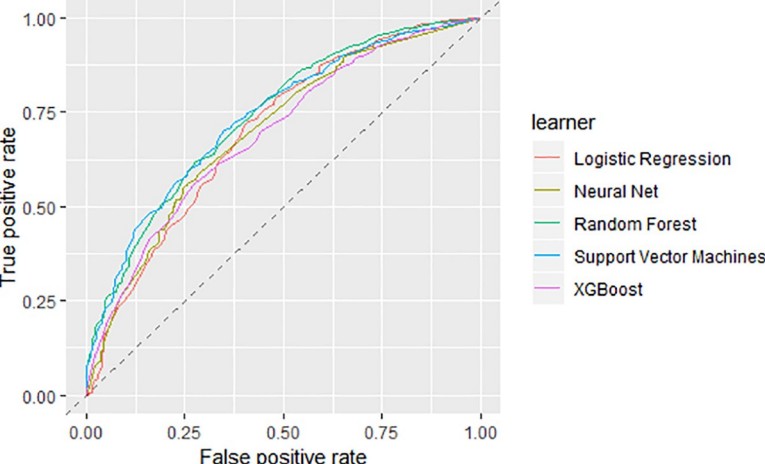

**Fig 3. AUC curves for test set.** The AUC performance for each technique using the test set.

## Variable importance

The variable importance for random forest revealed that the parent-reported mental health items ranked highly, as well as neighborhood quality, gestational age, and parity (Table 5). This indicates that model accuracy decreased significantly when these particular variables were permuted, i.e. randomly exchanged between individuals, during the analysis.

## Discussion

Using a large range of data from parent reports and register data from numerous Swedish national registers, this study predicted adolescent mental health reasonably well, with a maximum AUC of 0.739 on the test set (using the random forest model). Although the AUC indicates an adequate model, it is not accurate enough for clinical use. While the negative predictive value is at 96% indicates clinical level sensitivity, the positive predictive value of this model is only 15%. This indicates that only a small percentage of the children flagged will

**Table 5. Variable importance in random forest.**

| Predictor (Source) | Importance |
|---|---|
| Oppositional Defiant symptoms [1] | 136.97 |
| Impulsivity symptoms [1] | 94.05 |
| Inattention symptoms [1] | 92.66 |
| Executive dysfunction [1] | 87.72 |
| Emotional symptoms [1] | 76.82 |
| Neighborhood deprivation [2] | 64.03 |
| Peer difficulty [1] | 53.22 |
| Parity [3] | 44.17 |
| Gestational age at birth [3] | 43.71 |
| Separation anxiety [1] | 43.13 |

[1] Autism—Tics, AD/HD and other Comorbidities inventory [58]

[2] the Longitudinal Integration Database for Health Insurance and Labor Market Studies[35]

[3] Medical Birth Register [33]

actually reach our pre-specified cut-off for mental health problems, which should be compared to the prevalence in the sample of 10%.

The variable importance derived from the random forest model indicated that the model did not overly rely on any variable, thus the model would be relatively stable with the removal of any one variable, including those stable over time. The highest ranked variables were parent-reported mental health symptoms such as impulsivity, inattention, and emotional symptoms were important predictive factors for poor mental health at 15. Register information on neighborhood quality, parity and gestational age of birth were also deemed important. These findings fit within literature [17, 18, 44] and could potentially be used by clinicians, parents, or educators to identify at risk children for potential intervention.

The highest ranking variables were either parent-rated or could easily be reported by parents, this indicates that register information, which can be expensive or difficult for researchers to obtain, may not be necessary for a successful psychiatric risk model. Thus, future studies predicting adolescent mental health may want to place a greater emphasis on assessment from caregivers. Moreover, this provides further encouragement for parental involvement in clinicians' assessment of childhood and adolescent psychiatric prognosis and emotional well-being. Additionally, future studies with similar aims should focus on using symptom ratings for mental health, including neurodevelopmental disorders, for their model.

Sensitivity analysis showed that the model performance was slightly improved, although not significantly, with a more extreme cut-off (sensitivity analysis AUC = 0.765, 95% CI 0.698–0.826; random forest AUC = 0.739, 95% CI 0.708–0.769). This indicates that future studies can use cut-offs validated for their country or the original study based on preference. Additionally, this provides some evidence that the more extreme cases do not represent a distinct severe class.

In line with the No Free Lunch Theorem, all models performed with relatively similar accuracy [29]. A recent systematic review found no clear predictive performance advantage of using machine learning techniques instead of logistic regression, in a range of clinical prediction studies [45]. In our study, the similar performance to logistic regression may partially be attributed to the relatively linear associations from the predictors to the outcome, evident by the lack of significance for non-linear associations in our logistic regression model. When the data has a mostly linear relationship to the outcome, machine learning models will be very similar to standard regression [46]. Although random forest performed slightly better than the compared models, it may be unnecessary for studies with similar datasets and aims to use complex machine learning techniques instead of logistic regression when weighed against time spent learning the techniques, computational time, as well as interpretability of the model.

The strengths of this study include the comprehensive analysis of a wide variety of factors associated with adolescent mental health. Further, the use of parental reports indicates that these risk factors are identifiable by non-clinicians, indicating a low cost future solution for large scale mental health screens. The results need to be viewed in the light of several limitations. First, because we used a twin sample our findings may not be generalizable to singletons as our sample might have underlying differences in comparison to singletons. However, previous literature has found little difference in mental health between singletons and twins [47]. That said, zygosity did not rank as highly important, indicating that the model did not rely on the similarity between twins. On a similar note, our study results may not generalize outside of Sweden or Scandinavia, as all of our participants were Swedish born and we did not validate our results in an external sample. Second, the outcome as well as the most important variables were all parent-reported, this may have introduced an association due to a reporting bias. Additionally, because we used mixed data types (continuous, categorical, and binary) in our model it's possible that the variable importance could have been biased, however this effect is

likely to be mitigated as we did not sample with replacement [48]. Finally, the response rate between data collections was 55% [30], so it's likely that the nonresponders had elevated psychopathology symptoms compared to responders. Additionally, the performance of the model would likely improve with a larger sample size.

In summation, our models had a reasonable AUC, but no model had statistically significant higher performance than the other. Although supervised machine learning techniques are currently generating considerable interest across scientific fields, it may not be necessary for most studies to forgo logistic regression, especially for studies with smaller datasets featuring primarily linear relationships. Additionally, our results provide further support for diligent screening of neurodevelopmental symptoms and learning difficulties in children for later psychiatric vulnerabilities. Although, machine learning techniques seem to be promising for the integration of risks across different domains for the prediction of mental health problems in adolescence, it seems premature for implementation in clinical use. Nevertheless, as early treatment for these and other mental health symptoms has been found to largely mitigate negative outcomes and symptoms [49, 50], there is hope for prevention of negative mental health problems in adolescence with properly timed interventions.

## Supporting information

**S1 File. Variable codebook.** A list of variables considered for our model.
(XLSX)

**S1 Table. Support vector machine.** Optimal and explored parameters for the support vector machine model.
(DOCX)

**S2 Table. Neural network.** Optimal and explored parameters for the neural network model.
(DOCX)

**S3 Table. Random forest.** Optimal and explored parameters for the random forest model.
(DOCX)

**S4 Table. XGBoost.** Optimal and explored parameters for the XGBoost model.
(DOCX)

## Acknowledgments

The authors would like to thank Alexander Hatoum for his contribution to the code.

## Author Contributions

**Conceptualization:** Ashley E. Tate, Ralf Kuja-Halkola.

**Data curation:** Henrik Larsson, Sebastian Lundström, Paul Lichtenstein.

**Formal analysis:** Ashley E. Tate, Ryan C. McCabe, Ralf Kuja-Halkola.

**Funding acquisition:** Paul Lichtenstein.

**Methodology:** Ashley E. Tate, Ryan C. McCabe, Ralf Kuja-Halkola.

**Supervision:** Henrik Larsson, Sebastian Lundström, Paul Lichtenstein, Ralf Kuja-Halkola.

**Writing – original draft:** Ashley E. Tate, Ralf Kuja-Halkola.

**Writing – review & editing:** Ashley E. Tate, Ryan C. McCabe, Henrik Larsson, Sebastian Lundström, Paul Lichtenstein, Ralf Kuja-Halkola.

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
