## [Decision Letter · Decision Letter 0]

4 Dec 2019

PONE-D-19-24985

Predicting mental health problems in adolescence using machine learning techniques

PLOS ONE

Dear Ms Tate,

Thank you for submitting your manuscript to PLOS ONE. After careful consideration, we feel that it has merit but does not fully meet PLOS ONE’s publication criteria as it currently stands. Therefore, we invite you to submit a revised version of the manuscript that addresses the points raised during the review process.

We would appreciate receiving your revised manuscript by Jan 18 2020 11:59PM. To enhance the reproducibility of your results, we recommend that if applicable you deposit your laboratory protocols in protocols.io, where a protocol can be assigned its own identifier (DOI) such that it can be cited independently in the future. For instructions see: http://journals.plos.org/plosone/s/submission-guidelines#loc-laboratory-protocols

We look forward to receiving your revised manuscript.

Kind regards,

Parisa Rashidi

Academic Editor

PLOS ONE

Journal Requirements:

The Swedish Twin Registry is managed by Karolinska Institutet and receives funding through the Swedish Research Council under the grant no 2017-00641.

The Child and Adolescent Twin Study in Sweden study was supported by the Swedish Council for Working Life, funds under the ALF agreement, the Söderström Königska Foundation and the Swedish Research Council (Medicine, Humanities and Social Science; grant number 2017-02552, and SIMSAM). SL, PL

This work has received funding from the European Union’s Horizon 2020 Research and Innovation Programme under the Marie Sklodowska-Curie CAPICE Project grant agreement number 721567. (https://www.capice-project.eu/) AT, PL, SL

We acknowledge financial support from the Swedish Research Council for Health, Working Life and Welfare (project 2012-1678; PL), the Swedish Research Council (2016-01989; PL), as well as the the Swedish Initiative for Research on Microdata in the Social And Medical Sciences (SIMSAM) framework (340-2013-5867; PL)

I have read the journal's policy and the authors of this manuscript have the following competing interests:

H. Larsson has served as a speaker for Evolan Pharmaand Shire and has received research grants from Shire; all outside the submitted work. P. Lichtenstein has served as a speaker for Medice, also outside the submitted work.

R. McCabe serves as a data scientist for Spotify outside of the submitted work.

All other authors declare that no competing interests exist

Additional Editor Comments:

Based on the reviewers' comments, a minor revision is recommended for this manuscript. Please address reviewers' comments as appropriate.

Reviewers' comments:

Reviewer's Responses to Questions

**Comments to the Author**

1. Is the manuscript technically sound, and do the data support the conclusions?

Reviewer #1: Yes

Reviewer #2: Yes

2. Has the statistical analysis been performed appropriately and rigorously? 

Reviewer #1: Yes

Reviewer #2: I Don't Know

3. Have the authors made all data underlying the findings in their manuscript fully available?

Reviewer #1: No

Reviewer #2: No

4. Is the manuscript presented in an intelligible fashion and written in standard English?

Reviewer #1: Yes

Reviewer #2: Yes

5. Review Comments to the Author

Reviewer #1: Overall this is a very well-written paper with clear descriptions of motivation, methods, conclusions, and limitations. The authors were very thorough in describing variables and parameters used in the prediction models. I have only minor comments that I feel would improve clarity:

1. Typo in abstract: "METODS"  "METHODS"

2. The authors state that machine learning models are "black box", but typically this is in reference to deep learning models. Most of the models used in this paper would be considered conventional and "interpretable"

3. The authors state that CATSS participants are "described in detail elsewhere". This should at least be summarized in the current manuscript.

4. Was there any particular reason that 50% was the missingness threshold for removing variables? I think it would be nice to examine possible missingness patterns, e.g. particular variables missing for certain subgroups.

5. Since the described models are not computationally expensive, it might be nice to perform nested cross validation as opposed to a fixed train/val/test split.

6. The class imbalance should be mentioned in the main manuscript.

7. Table 2 should be referenced for the following line: "Descriptive statistics were created for each set to determine the quality of the partition"

8. Authors should slightly reword the description of training procedure. I assume "fit was determined by finding the maximum AUC" is referring to AUC on the tune set, but this should be explicitly mentioned. It almost reads like models were first trained on the training set before moving on to the tune set, but both of these should be used simultaneously in the cross validation procedure.

9. Why weren't feature importances explored for models like logistic regression or SVM? It is certainly possible.

10. I assume the "best performing model" is based on tune set performance (and not test set), but this should be explicitly mentioned.

11. I find the description of the neural network to be problematic, several important parameters were not mentioned (# of layers, optimizer and its parameters, dropout, etc.). Furthermore, the final hidden dimension of 3 seems very low.

12. The authors should construct a supplemental table of the ranges of parameters explored in the random search.

Reviewer #2: This is a study of predictors of mental health issues in a sample of 7,638 Swedish twins. Predictors were collected on them at ages 9-12 and the mental health criterion data were collected at age 15. Although governmental data on Swedish twins is used in this study, the fact that they are twins is irrelevant and appears to pose no source of bias regarding the results. Of 474 variables collected from various governmental data sources, 85 survived scientific scrutiny and were included in the machine learning and regression models reported. Findings suggested that both kinds of analyses produced AUC scores above .7. Apparently these values are not adequate for clinical application, but they are certainly informative for behavioral scientists. Two very important findings from this investigation are (a) logistic regression was adequate for this work, so machine learning analyses may be unnecessary in similar future studies, and (b) the most powerful predictors of mental health issues among these Swedish teens came from parent reports, which are far faster and easier data to collect than most of the other predictors. These two findings are important to share because they provide a green light to the work of investigators in this area who may not be proficient in machine learning and who may only have access to data from parents. In addition, based on these findings, extramural funders may seek to fund these more affordable projects, instead of rejecting them in favor of funding projects that use more costly machine learning analyses (with the need for a lot of data) and governmental data sources.

I am not a machine learning expert, so I cannot speak to the statistical conclusion validity of those analyses, but I am competent in logistic regression and saw no issues in those analyses.

In several places, the authors need to be careful not to elevate or hint at elevating nonsignificant effects to significance. When the authors say two values are different, then later say they are not significantly different, they blur the conclusion. Statistically the numbers are not different. Better simply to say the two values did not significantly differ and leave it at that. I am no fan of null hypothesis statistical testing, but the authors chose that approach, and some scientific communities still use that approach, so the authors need to remain true to that approach, which posits that findings either are or are not different based on statistical significance.

Because participants being twins was immaterial to the scientific questions addressed in this study, the authors should explain why they used a twin sample. It seems as though they could have gotten data on a far larger sample of Swedish children if they did not restrict their focus to twins, who on average represent less than 3% of a population. It could be that the kinds of data collected on Swedish twins simply are not collected on their non-twin counterparts. If that's so, the authors should say that.

The ms. would be improved by a section that very specifically enumerated important next steps in predicting teen mental health issues, given these twin data. What do the authors think would be good ways for scholars to increase the AOC to levels appropriate for clinical use, for example? Other scholars would very much appreciate this kind of insight to guide their work.

An important limit to this work is cultural. Findings based on Swedes and Swedish culture may not broadly generalize, especially with regard to outcomes as socially defined as mental health concerns. So in addition to the five limitations briefly included in the Discussion, I suggest the authors add concerns about generalizability beyond Sweden and other very similar and similarly homogeneous nations.

6. PLOS authors have the option to publish the peer review history of their article (what does this mean?). If published, this will include your full peer review and any attached files.

Reviewer #1: No

Reviewer #2: No

---

## [Author Response · Author response to Decision Letter 0]

23 Jan 2020

Dear Editor,

First, we would like to thank you for your timely feedback and thorough review of the paper. Our response can be read in the following key:

Reviewer’s comment Revisions Response 

Lines numbers correspond to the document without tracked changes

Journal Comment 1. When submitting your revision, we need you to address these additional requirements.

Answer 1: Thank you for bringing this to our attention. We have unbolded the title and additionally have added a new line for the corresponding author’s email. We have also changed the supporting files names to add the file type in the name, e.g. “Fig1” -> “Fig1.tif”. Please let us know if any additional style requirements were unintentionally omitted.

Journal Comment 2. Thank you for stating the following in the Acknowledgments Section of your manuscript:

The Swedish Twin Registry is managed by Karolinska Institutet and receives funding through the Swedish Research Council under the grant no 2017-00641.

The Child and Adolescent Twin Study in Sweden study was supported by the Swedish Council for Working Life, funds under the ALF agreement, the Söderström Königska Foundation and the Swedish Research Council (Medicine, Humanities and Social Science; grant number 2017-02552, and SIMSAM). SL, PL

This work has received funding from the European Union’s Horizon 2020 Research and Innovation Programme under the Marie Sklodowska-Curie CAPICE Project grant agreement number 721567. (https://www.capice-project.eu/) AT, PL, SL

We acknowledge financial support from the Swedish Research Council for Health, Working Life and Welfare (project 2012-1678; PL), the Swedish Research Council (2016-01989; PL), as well as the the Swedish Initiative for Research on Microdata in the Social And Medical Sciences (SIMSAM) framework (340-2013-5867; PL)

Answer 2. We have removed the funding information from lines 296 – 298. We would like to kindly request that our funding statement be updated to include the lines:

We acknowledge The Swedish Twin Registry for access to data. The Swedish Twin Registry is managed by Karolinska Institutet and receives funding through the Swedish Research Council under the grant no 2017-00641.

Journal Comment 3. Thank you for stating the following in the Competing Interests section:

I have read the journal's policy and the authors of this manuscript have the following competing interests:

H. Larsson has served as a speaker for Evolan Pharmaand Shire and has received research grants from Shire; all outside the submitted work. P. Lichtenstein has served as a speaker for Medice, also outside the submitted work.

R. McCabe serves as a data scientist for Spotify outside of the submitted work.

All other authors declare that no competing interests exist

Answer 3. Thank you for bringing this to our attention. We have added included an updated version of the competing interests to our cover letter.

Journal Comment 4. We note that you have indicated that data from this study are available upon request. PLOS only allows data to be available upon request if there are legal or ethical restrictions on sharing data publicly. For more information on unacceptable data access restrictions, please see http://journals.plos.org/plosone/s/data-availability#loc-unacceptable-data-access-restrictions.

Answer 4. Thank you for addressing this. We have added the following response to this prompt in our cover letter:

Moreover, we regret to add that we are unable to share even de-identified data, as legally bound by the Swedish Serecy Act. Data from the national Swedish registers and twin register were used for this study and made available by ethical approval. The data used for this study include: Swedish Twin Registry, National Patient Register, Multi-Generation Register, Medical Birth Register, Prescribed Drug Register, the Longitudinal Integration Database for Health Insurance and Labor Market Studies. Researchers may apply for access these data sources through the Swedish Research Ethics Boards (etikprovningsmyndigheten.se; kansli@cepn.se) and from the primary data owners: Swedish Twin Registry (str-research@meb.ki.se), Statistics Sweden (scb@scb.se), and the National Board of Health and Welfare (socialstyrelsen@socialstyrelsen.se), in accordance with Swedish law.

Reviewer #1: Overall this is a very well-written paper with clear descriptions of motivation, methods, conclusions, and limitations. The authors were very thorough in describing variables and parameters used in the prediction models. I have only minor comments that I feel would improve clarity:

The authors would like to sincerely thank you for the kind evaluation and encouragement on the manuscript.

Question 1: Typo in abstract: "METODS"  "METHODS"

Answer 1: The typo in line 34 has been corrected.

Question 2: The authors state that machine learning models are "black box", but typically this is in reference to deep learning models. Most of the models used in this paper would be considered conventional and "interpretable"

Answer 2: Thank you for this comment, we agree that the reference to “black box” was misplaced. We have changed the text accordingly, lines 92 – 94 have been changed to:

Beyond their proven efficacy, tree based models provide information on how extensively a variable was used for the model, or variable importance, which gives some insight to the models’ classification process.

Question 3: The authors state that CATSS participants are "described in detail elsewhere". This should at least be summarized in the current manuscript.

Answer 3: Additional information has been added on lines 104 – 106:

Participants came from the Child and Adolescent Twin Study in Sweden (CATSS), an ongoing, longitudinal study containing 15,156 twin pairs born in Sweden. During the first wave, the twins’ parents were contacted close to their 9th or 12th birthdays for a phone interview, this wave had a response rate of 80% (36), while the second wave at age 15 had a response rate of ~55%.

Question 4: Was there any particular reason that 50% was the missingness threshold for removing variables? I think it would be nice to examine possible missingness patterns, e.g. particular variables missing for certain subgroups.

Answer 4: Thank you for this comment. The 50% missingness was chosen a bit arbitrarily, the main aim was to keep variables with acceptable coverage in the prediction model and we felt that overly imputed variables ultimately would not lead to better results compared to removing them. CATSS is an ongoing longitudinal study and some of questions were changed or added over the years. This means that some questions had a high rate of missingness because only a small percentage of our sample was asked. Additionally, there were several gated questions that also had a high degree of missingness. Thus, our approach automatically excludes these questions. 

A distribution of the missingness in our data is visualized in the below figure:

As can be seen, the cut-off of 50% missingness (chosen a priori) removes a set of variables with 90% or more missingness and a set of variables with 70-80% missingness (more borderline quality of variables). We believe our choice, albeit somewhat arbitrary, achieves a good balance between retaining variables with sufficient coverage/quality while not being overly conservative.

Question 5; Since the described models are not computationally expensive, it might be nice to perform nested cross validation as opposed to a fixed train/val/test split.

Answer 5: This is a great topic for discussion, thank you for bringing this up. One problem with this approach would be the potential splitting of twins between the subsets in nested-cross validation, which could lead to overfitting. Without the tune set as a “safety net” we thought it would be hard to catch the potentiality for overfitting before moving to the test set. This concerned the authors as we wanted to avoid training a new model after moving to the test set. To our knowledge, no such control for this issue within nested-cross validation exists within the common ML packages in R (please let us know if you know of a potential solution). Notably, caret has groupKfolds, but in practice this did not turn out to be helpful [1]. 

We did additional sensitivity analysis by training a random forest model with nested cross validation. We split our data into a train and test set (70/30 split) and found similar performance to our previously created models (AUC=0.743, 95% CI 0.712 - 0.773) compared to our top performing model (AUC=0.739, 95% CI 0.708 – 0.769). This result indicates that the tuning approach comes down to personal preference, although the authors would like to contend that nested cross validation seems to be a more streamlined option. 

Question 6: The class imbalance should be mentioned in the main manuscript.

Answer 6: Thank you for this suggestion, we agree and an additional sentence has been added to the results section on lines 178 – 180:

Our classes were fairly imbalanced as only 12% of our sample reached the cut off, we mitigated the effects of this through a combination of over- and under sampling on the training set using SMOTEBoost.

Question 7: Table 2 should be referenced for the following line: "Descriptive statistics were created for each set to determine the quality of the partition"

Answer 7: The suggested edit has been made on line 147.

Question 8: Authors should slightly reword the description of training procedure. I assume "fit was determined by finding the maximum AUC" is referring to AUC on the tune set, but this should be explicitly mentioned. It almost reads like models were first trained on the training set before moving on to the tune set, but both of these should be used simultaneously in the cross validation procedure.

Answer 8: Thank you for spotting this omission. Edits were made on lines 154 – 160:

We created prediction models using several machine learning techniques: random forest, XGBoost, logistic regression, neural network and support vector machines (Table 1) to determine which produced the best fitting model for a test set. Using cross validation, each technique trained multiple models using the training set and tested their performance on a subset of the training set. The model with the lowest error was then tested using the tune set. Once the performance in the tune set was deemed satisfactory, the final models were then fitted to the test set.

Question 9: Why weren't feature importances explored for models like logistic regression or SVM? It is certainly possible.

Answer 9: While possible, we felt that this was not worth delving into for two reasons: the non-superiority of any one model, and the feature importances showed similar results to the random forest model (parent reported mental health symptoms ranked highest). Since random forest was the slightly better model we chose to only report the variable importance for that model.

Question 10: I assume the "best performing model" is based on tune set performance (and not test set), but this should be explicitly mentioned.

Answer 10: Thank you for letting us clarify this. The set we are referring to is in fact the test set. 

This has been clarified on lines 154 – 155:

We created prediction models using several machine learning techniques: random forest, XGBoost, logistic regression, neural network and support vector machines (Table 1) to determine which produced the best fitting model for a test set.

Additionally on lines 224 – 226:

Using a large range of data from parent reports and register data from numerous Swedish national registers, this study predicted adolescent mental health reasonably well, with a maximum AUC of 0.739 on the test set (using the random forest model).

Question 11: I find the description of the neural network to be problematic, several important parameters were not mentioned (# of layers, optimizer and its parameters, dropout, etc.). Furthermore, the final hidden dimension of 3 seems very low.

Answer 11: Thank you for your feedback, this particular portion of the analysis gave the authors the most trouble! The parameters not mentioned were not adjusted or modified in our analysis. Given our relatively linear data and small number of participants (from an ML standpoint), a larger number of hidden dimensions were unnecessary [2], moreover we tried a range of hidden dimensions and 3 was determined to lead to the best fitting model. See Answer 12 below for further information about hyper-parameters tested.

Question 12: The authors should construct a supplemental table of the ranges of parameters explored in the random search.

Answer 12: The suggested edit has been added to tables S1 – S4 in the supporting information.

This edit has also been reflected on line 191 – 192:

A full list of the optimal parameters and the ranges tried for each model can be found in S1-S4 tables.

Reviewer #2: This is a study of predictors of mental health issues in a sample of 7,638 Swedish twins. Predictors were collected on them at ages 9-12 and the mental health criterion data were collected at age 15. Although governmental data on Swedish twins is used in this study, the fact that they are twins is irrelevant and appears to pose no source of bias regarding the results. Of 474 variables collected from various governmental data sources, 85 survived scientific scrutiny and were included in the machine learning and regression models reported. Findings suggested that both kinds of analyses produced AUC scores above .7. Apparently these values are not adequate for clinical application, but they are certainly informative for behavioral scientists. Two very important findings from this investigation are (a) logistic regression was adequate for this work, so machine learning analyses may be unnecessary in similar future studies, and (b) the most powerful predictors of mental health issues among these Swedish teens came from parent reports, which are far faster and easier data to collect than most of the other predictors. These two findings are important to share because they provide a green light to the work of investigators in this area who may not be proficient in machine learning and who may only have access to data from parents. In addition, based on these findings, extramural funders may seek to fund these more affordable projects, instead of rejecting them in favor of funding projects that use more costly machine learning analyses (with the need for a lot of data) and governmental data sources.

I am not a machine learning expert, so I cannot speak to the statistical conclusion validity of those analyses, but I am competent in logistic regression and saw no issues in those analyses.

Question 13: In several places, the authors need to be careful not to elevate or hint at elevating nonsignificant effects to significance. When the authors say two values are different, then later say they are not significantly different, they blur the conclusion. Statistically the numbers are not different. Better simply to say the two values did not significantly differ and leave it at that. I am no fan of null hypothesis statistical testing, but the authors chose that approach, and some scientific communities still use that approach, so the authors need to remain true to that approach, which posits that findings either are or are not different based on statistical significance.

Answer 13: Thank you for this valuable comment, we agree that a clearer distinction needed to be made. We’ve edited several lines to further clarify that there were no significant differences between the models.

Line 224 – 226:

Using a large range of data from parent reports and register data from numerous Swedish national registers, this study predicted adolescent mental health reasonably well, with a maximum AUC of 0.739 on the test set (using the random forest model).

In line 248- 249 this sentence was deleted:

The model created with random forest closely followed by support vector machine had the highest AUCs in the test set. However,

Additionally line 281 – 282 was changed to:

In summation, our models had a reasonable AUC, but no model had statistically significant higher performance than the other.

Question 14: Because participants being twins was immaterial to the scientific questions addressed in this study, the authors should explain why they used a twin sample. It seems as though they could have gotten data on a far larger sample of Swedish children if they did not restrict their focus to twins, who on average represent less than 3% of a population. It could be that the kinds of data collected on Swedish twins simply are not collected on their non-twin counterparts. If that's so, the authors should say that.

Answer 14: Thank you for this comment, this is indeed the case, the depth of information, e.g. longitudinal questionnaires, obtained from the Child and Adolescent Twin Study in Sweden is simply not available in singleton samples easily accessible by the authors.

An edit at lines 107 - 108 has been added:

This sample population was chosen due to the depth of information available, including questionnaire and register data

Question 15: The ms. would be improved by a section that very specifically enumerated important next steps in predicting teen mental health issues, given these twin data. What do the authors think would be good ways for scholars to increase the AOC to levels appropriate for clinical use, for example? Other scholars would very much appreciate this kind of insight to guide their work.

Answer 15: Thank you for this suggestion. We’ve added this information to lines 245 - 247:

Additionally, future studies with similar aims should focus on using symptom ratings for mental health, including neurodevelopmental disorders, for their model.

Question 16: An important limit to this work is cultural. Findings based on Swedes and Swedish culture may not broadly generalize, especially with regard to outcomes as socially defined as mental health concerns. So in addition to the five limitations briefly included in the Discussion, I suggest the authors add concerns about generalizability beyond Sweden and other very similar and similarly homogeneous nations.

Answer 16: We’ve added this limitation to lines 272 – 273:

On a similar note, our study results may not generalize outside of Sweden or Scandinavia, as all of our participants were Swedish born and we did not validate our results in an external sample.

References

1. Kuhn M. Caret: classification and regression training. Astrophysics Source Code Library 2015 

2. Dreiseitl S, Ohno-Machado L. Logistic regression and artificial neural network classification models: a methodology review. Journal of biomedical informatics 2002;35(5-6):352-59

---

## [Decision Letter · Decision Letter 1]

28 Feb 2020

Predicting mental health problems in adolescence using machine learning techniques

PONE-D-19-24985R1

Dear Dr. Tate,

We are pleased to inform you that your manuscript has been judged scientifically suitable for publication and will be formally accepted for publication once it complies with all outstanding technical requirements.

With kind regards,

Wajid Mumtaz

Academic Editor

PLOS ONE

Additional Editor Comments (optional):

Reviewers' comments:

Reviewer's Responses to Questions

**Comments to the Author**

1. If the authors have adequately addressed your comments raised in a previous round of review and you feel that this manuscript is now acceptable for publication, you may indicate that here to bypass the “Comments to the Author” section, enter your conflict of interest statement in the “Confidential to Editor” section, and submit your "Accept" recommendation.

Reviewer #1: All comments have been addressed

2. Is the manuscript technically sound, and do the data support the conclusions?

Reviewer #1: Yes

3. Has the statistical analysis been performed appropriately and rigorously? 

Reviewer #1: Yes

4. Have the authors made all data underlying the findings in their manuscript fully available?

Reviewer #1: No

5. Is the manuscript presented in an intelligible fashion and written in standard English?

Reviewer #1: Yes

6. Review Comments to the Author

Reviewer #1: The authors have fully addressed my concerns and made satisfactory improvements to the revised manuscript.

7. PLOS authors have the option to publish the peer review history of their article (what does this mean?). If published, this will include your full peer review and any attached files.

Reviewer #1: No

---

## [Editor Report · Acceptance letter]

3 Mar 2020

PONE-D-19-24985R1 

Predicting mental health problems in adolescence using machine learning techniques 

Dear Dr. Tate:

I am pleased to inform you that your manuscript has been deemed suitable for publication in PLOS ONE. Congratulations! Your manuscript is now with our production department. 

With kind regards,

on behalf of

Dr. Wajid Mumtaz 

Academic Editor

PLOS ONE